# DNA methylation in mice is influenced by genetics as well as sex and life experience

Sara A. Grimm[1], Takashi Shimbo[1], Motoki Takaku[1], James W. Thomas[2], Scott Auerbach[3], Brian D. Bennett[1], John R. Bucher[3], Adam B. Burkholder[1], Frank Day[1], Ying Du[1], Christopher G. Duncan[1], John E. French[3], Julie F. Foley[3], Jianying Li[1], B. Alex Merrick[3], Raymond R. Tice[3], Tianyuan Wang[1], Xiaojiang Xu[1], NISC Comparative Sequencing Program, Pierre R. Bushel[1], David C. Fargo[1], James C. Mullikin [2] & Paul A. Wade [1]

DNA methylation is an essential epigenetic process in mammals, intimately involved in gene regulation. Here we address the extent to which genetics, sex, and pregnancy influence genomic DNA methylation by intercrossing 2 inbred mouse strains, C57BL/6N and C3H/HeN, and analyzing DNA methylation in parents and offspring using whole-genome bisulfite sequencing. Differential methylation across genotype is detected at thousands of loci and is preserved on parental alleles in offspring. In comparison of autosomal DNA methylation patterns across sex, hundreds of differentially methylated regions are detected. Comparison of animals with different histories of pregnancy within our study reveals a CpG methylation pattern that is restricted to female animals that had borne offspring. Collectively, our results demonstrate the stability of CpG methylation across generations, clarify the interplay of epigenetics with genetics and sex, and suggest that CpG methylation may serve as an epigenetic record of life events in somatic tissues at loci whose expression is linked to the relevant biology.

[1] Division of Intramural Research, NIEHS, 111 TW Alexander Drive, Research Triangle Park, NC 27709, USA. [2] NIH Intramural Sequencing Center, National Human Genome Research Institute, 5625 Fishers Lane, Rockville, MD 20852, USA. [3] National Toxicology Program, NIEHS, 111 TW Alexander Drive, Research Triangle Park, NC 27709, USA. Correspondence and requests for materials should be addressed to P.A.W. (email: wadep2@niehs.nih.gov)

Methylation of cytosine in the context of the simple palindromic dinucleotide 5′ CG 3′ represents the most common form of DNA modification in mammals[1,2]. Maintenance of DNA methylation states following DNA replication constitutes an essential mechanism wherein daughter cells inherit cell-type specific epigenetic programs. The global pattern of DNA methylation is reprogrammed during genesis of germ cells and also during very early embryogenesis, establishing a common epigenetic slate for development and differentiation[3], raising questions regarding the extent to which DNA methylation patterns in offspring resemble those in parents. Nonetheless, evidence exists that DNA methylation patterns may be, to some extent, under genetic control[4–6], suggesting a mechanistic basis for similarity between parents and offspring.

The relationship between local DNA methylation and transcription factor–DNA interactions appears to be complex. Biochemical and genomic analyses have defined multiple transcription factors whose productive interaction with local DNA sequence is blocked by cytosine methylation that occurs within cognate recognition sequences[7–11], and other transcription factors whose binding is facilitated by DNA methylation[11–13]. So-called pioneer transcription factors are widely believed to have the inherent capacity to penetrate local chromatin-based barriers to binding, giving them the capacity to direct alterations in cell identity[14]. Further, transcription factor binding has been posited as a mechanism wherein local CpG dinucleotides are protected from action of DNA methyltransferases, leading to local hypomethylation[15–19]. These observations suggest that different transcription factors may influence, or be influenced by, local DNA methylation patterns in different ways. The downstream output of gene transcription is also likely to be influenced in a complex manner dependent on rate-limiting transcription factors.

Here, we address the relationship of DNA methylation patterns in somatic tissue across generation using inbred mouse strains in a genetic model system. Our findings demonstrate thousands of local sites where different strains of inbred mice, grown in identical conditions, differ in DNA methylation pattern. These genotype-dependent differences in local DNA methylation are preserved on parental alleles in hybrid F1 progeny, suggesting linkage to DNA sequence. We suggest that the linkage of DNA methylation state to DNA sequence results, in part, from its relationship to transcription factor biology. In some cases, genetic control of transcription factor binding correlates with differential methylation in our genetic system, as observed for other epigenetic marks[20–22] and as has been reported for DNA methylation[16–18,23,24]. In other cases, it seems likely that local DNA methylation influences the quality of transcription factor interaction with local DNA sequences, either in a positive or negative manner. Furthermore, in comparison of animals of different sex and life history, we find that major life events such as pregnancy may leave a DNA methylation signature in nonreproductive somatic tissues at loci whose expression is linked to the relevant biology.

## Results

**A genetic system for study of DNA methylation.** To address the degree of similarity of DNA methylation patterns in a somatic tissue when comparing parents to offspring, we crossed C57BL/6N and C3H/HeN mice (subsequently referred to as B6 and C3) in both directions to derive offspring (both male and female F1s) from a total of six crosses (three of each type). Animals were reared in a controlled environment and were sacrificed at identical ages to minimize confounding variables. We chose liver as a somatic tissue of interest and prepared genomic DNA for further analysis. We performed whole-genome sequencing (30–35×

genome coverage, Supplementary Data 1) and constructed local genome assemblies for B6 and C3, from which we identified approximately 2.8 million autosomal single-nucleotide variations (SNVs) from the reference (mm9) genome.

Shotgun whole-genome bisulfite sequencing was performed (see Methods) on all 24 animals, collecting approximately $1.8 \times 10^{12}$ nucleotides after filtering, mapping and deduplication (Supplementary Data 1). This data set represents total read depth of approximately 150× genome coverage each for B6, C3, and the hybrid F1 progeny, B6C3F1 and C3B6F1 (Supplementary Data 1; Supplementary Figure 1). Global methylation levels were similar to previously reported values[1,2], (Supplementary Figure 1, Supplementary Table 1, Supplementary Table 2). In the strains utilized for this study, we identified 18.8 million autosomal CpG dinucleotides that were conserved in both our local genomes as well as the reference genome; we limited all further analysis to these 18.8 million CpGs.

**Methylation variability at the individual CpG level.** As one goal of our study was to compare CpG methylation patterns across groups of animals, we began our analysis by assessing the extent of gross methylation variability across the animals in our study. We calculated the mean and standard deviation of the percent of methylated alleles within biological replicate groups (e.g., B6 animals) for 20,000 randomly selected CpGs with different coverage levels (Fig. 1a). CpG sites with less than 10× coverage had a mean standard deviation of slightly greater than 10%. This value declined with increasing read depth, approaching a limit of approximately 5% (Fig. 1a). Coverage depth beyond 11–20× coverage provided minimal improvement in standard deviation, suggesting that this level of variability is an inherent property of the biological system or is technical in nature.

Even accounting for CpG sites lacking sufficient sequencing depth, there exist outliers with high standard deviation, suggesting a subset of CpGs may be inherently variable in methylation level across the population of alleles sampled. We sought to define the number of such outliers by plotting standard deviation (SD) across all animals (Fig. 1b). While approximately 95% of all CpG sites had SD less than 10%, a subset of CpGs were more highly variable (Fig. 1b). These CpGs are located in regions with higher CpG density than randomly selected controls, have intermediate levels of CpG methylation, and tend to be excluded from CpG islands (Fig. 1c), similar to previous reports[25]. While this set of CpGs may have some unique biology, we could not identify a logical rationale for their exclusion from subsequent analyses.

**Differentially methylated regions by genotype.** To assess the impact of genetics on DNA methylation patterning, we sought to identify regions where methylation differed between parents, to ask whether the pattern in F1 offspring resembled one parent or differed from both. Currently, there is not a consensus statistical method for identification of differentially methylated genomic regions in whole-genome bisulfite sequencing data. Therefore, we employed two different approaches which use fundamentally different strategies. DSS uses a beta-binomial hierarchical model with a parametric statistical test[26]. Metilene[27] employs a scoring method to identify maximal between-group methylation differences in genomic regions of minimum length in combination with a nonparametric test. After employment of each tool, we refined the results with the following filters: (1) minimum methylation difference > 20%, (2) minimum of five CpG sites per DMR, (3) disregard any DMRs where one group has read depth > 500×, and (4) disregard any DMRs where the ratio of average

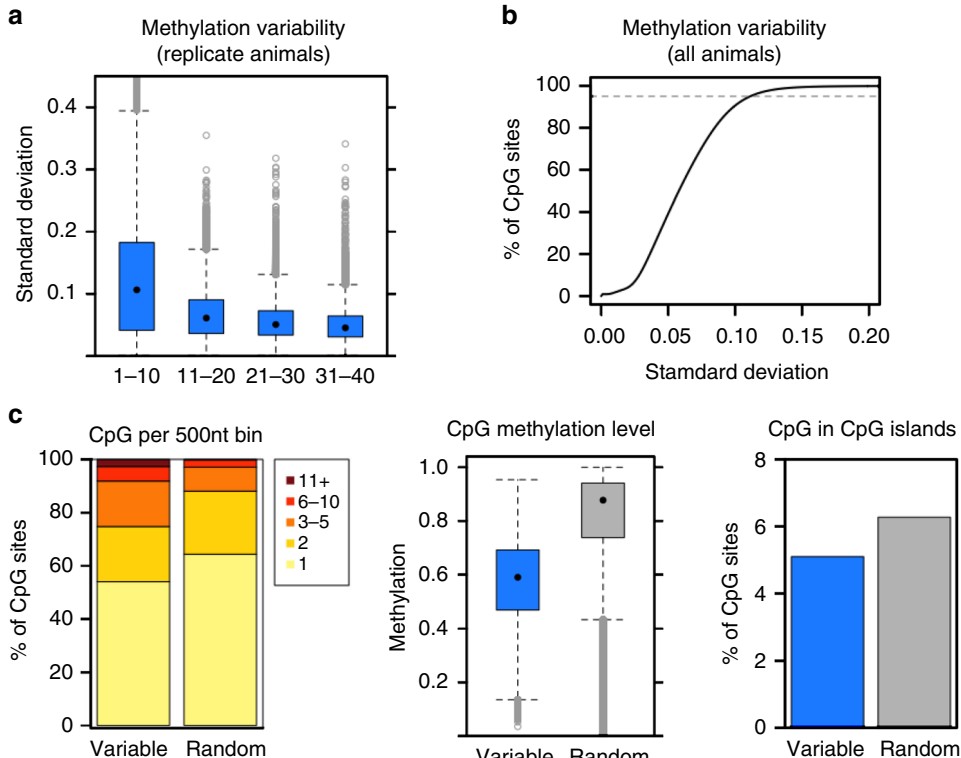

**Fig. 1** Variability in methylation observed across animals. **a**. Standard deviation of methylation level among biological replicates. CpG sites are binned according to site depth, where all replicates (three males and three females) must have depth within the specified range (e.g., 1–10 and 11-20). Totally, 20,000 CpG sites per replicate group (genotype) per depth bin were randomly selected for the boxplot. **b** Standard deviation of methylation level among all animals shown as a cumulative distribution function. Per-site calculations include only animals with depth at least 10; CpG sites are excluded if most animals lack sufficient depth (at least 18 of the 24 study animals must have depth > 10 at the CpG site). The top fifth percentile (most variable sites) is indicated by a dashed gray line. **c** Distribution of selected sites in 500 nt tiled windows (left), methylation level of selected sites (middle), and overlap of selected sites with CpG islands (right). Variable refer to sites among top 5% of standard deviation among all animals as in panel B (N = 871,166); random refer to another 5% of sites (N = 871,166) meeting the same depth criteria. Color scale in left panel indicates the number of CpG sites per 500nt tiled window. In the box-and-whisker plots, the box depicts the 25th to 75th percentiles, the black dot is the median, the whiskers extend to data points up to 1.5*IQR beyond the box, and open gray circles are data points outside the whisker range

depth between comparison groups is >5. We provide results derived from both methods throughout this report.

We initially compared CpG methylation in all six B6 parental animals, regardless of sex, to all six C3 parental animals. Each computational method successfully identified differentially methylated CpGs (DMCs). DSS identified 72,275 DMCs representing approximately 0.4% of CpG sites analyzed while Metilene called 185,194 DMCs (approximately, 1% of CpG sites analyzed). DSS identified a total of 6380 autosomal DMRs covering approximately 3.5 million bp (approximately, 0.1 % of the genome); Metilene identified 2569 DMRs covering 1 million bp (approximately, 0.04 % of the genome) (Table 1). Both computational methods called DMRs of each polarity (B6 methylation level higher than C3 and vice versa) at equivalent rates (Supplementary Data 2). While the two different computational methods each identified unique DMRs (Supplementary Figure 2A), the degree of overlap between methods was striking, with most DMRs called by Metilene also called by DSS (Supplementary Figure 2B).

Exemplar DMRs are illustrated in genome browser format for each computational method (Fig. 2a; Supplementary Figure 3A). The DMRs identified by both methods are relatively short genomic intervals; the vast majority are less than 1 kb in size (Fig. 2b; Supplementary Figure 3B). When we assessed the relationship of DMRs to genes, we found that DMRs are far from promoters; roughly 70% of all DMRs are more than 10 kb away

from the nearest RefSeq transcription start site (Fig. 2c; Supplementary Figure 3C). Consistent with the minimum methylation difference specified as a parameter in DMR calling, most DMRs have a substantial difference in methylation between the two strains (Fig. 2d; Supplementary Figure 3D). CpG content in DMRs was consistently higher than in random genomic intervals (Fig. 2e; Supplementary Figure 3E) although it did not reach the CpG density characteristic of CpG islands[28,29]. We compared the DMRs with existing data on histone modifications and nuclease accessibility, finding that about 20% of DMRs overlap with DNase hypersensitive sites as measured in C57BL/6J liver[30] (Fig. 2f; Supplementary Figure 3F), suggesting overlap with regulatory DNA. Further, nearly one-half of all DMRs also overlap (Fig. 2f; Supplementary Figure 3F) with a known enhancer histone mark, monomethylation of lysine 4 of histone H3 (H3K4me1). These data suggest that genotype–DMRs overlap with regulatory DNA, most likely at enhancers, a pattern observed in other types of comparisons[31,32].

**Parental DNA methylation is recapitulated in progeny.** DNA methylation scores[33] at the DMRs identified by comparison of B6 vs. C3 parental animals were calculated and visualized by hierarchical clustering of all study animals including progeny (Fig. 3a; Supplementary Figure 4A). The study animals cluster by genotype, with hybrid F1 animals occupying a position intermediate

**Table 1 Characteristics of differential methylation**

| Comparison | Tool | DMCs | DMRs | DMRs by type | DMCs in DMRs | % DMCs in DMRs | % DMR CpGs that are DMCs | DMR % of autosomes |
|---|---|---|---|---|---|---|---|---|
| B6 vs. C3 | DSS | 72,275 | 6380 | B6 > C3 3166 C3 > B6 3214 | 47,029 | 65.1 | 65.5 | 0.143 |
|  | Metilene | 185,194 | 2569 | B6 > C3 1262 C3 > B6 1307 | 23,839 | 12.9 | 78.8 | 0.042 |
| Male vs. female (B6 + C3 animals) | DSS | 12,224 | 1575 | M > F 534 F > M 1041 | 8532 | 69.8 | 58.4 | 0.036 |
|  | Metilene | 51,675 | 439 | M > F 172 F > M 267 | 3426 | 6.6 | 81.1 | 0.008 |
| Male vs. female (B6 animals) | DSS | 7400 | 1083 | M > F 415 F > M 668 | 5108 | 69.0 | 51.8 | 0.020 |
|  | Metilene | 37,327 | 70 | M > F 25 F > M 45 | 653 | 1.8 | 54.6 | 0.002 |
| Male vs. female (C3 animals) | DSS | 11,142 | 1539 | M > F 548 F > M 991 | 7495 | 67.3 | 53.1 | 0.029 |
|  | Metilene | 62,785 | 93 | M > F 29 F > M 64 | 1018 | 1.6 | 64.4 | 0.003 |
| Male vs. female (F1 animals) | DSS | 6007 | 720 | M > F 65 F > M 655 | 4053 | 67.5 | 59.1 | 0.017 |
|  | Metilene | 32,574 | 207 | M > F 26 F > M 181 | 1655 | 5.1 | 83.2 | 0.004 |
| Parental female vs. virgin female | DSS | 2257 | 305 | p > v 10 v > p 295 | 1467 | 65.0 | 51.7 | 0.006 |
|  | Metilene | 17,798 | 68 | p > v 2 v > p 66 | 437 | 2.5 | 61.8 | 0.001 |

between the parental strains. The DNA methylation levels were visualized across all animals by calculating the average weighted methylation score for each DMR, binning the data into quartiles, and visualizing as box and whisker plots (Fig. 3b; Supplementary Figure 4B). In all cases, DNA methylation level in the F1 progeny, regardless of the direction of the genetic cross, is intermediate between the parental levels. The simplest explanation for these results is that F1 progeny recapitulate DNA methylation patterns in *cis* on alleles inherited from their respective parents. To directly visualize allele specificity of DNA methylation in F1 progeny at select DMRs, we examined individual bisulfite reads where a DMR CpG and a diagnostic sequence difference between strains fell on the same sequence read (or read pair). We found that the DNA methylation level of parental alleles in F1 animals appears nearly identical to that found in the parents in the overwhelming majority of cases; exemplar DMRs are depicted (Fig. 3c; Supplementary Figure 4C). These data demonstrate that DNA methylation patterns are recreated on respective parental alleles in somatic tissue in offspring regardless of which germ line they pass through.

Recapitulation of methylation in an allele-specific manner implies a linkage of methylation pattern to genetic information. Accordingly, we assessed the relationship of DMRs with strain-specific differences in DNA sequence. DMRs, when compared to random genomic intervals of the same size, have a higher frequency of local SNVs (Fig. 3d; Supplementary Figure 4D). Further DMRs that do not contain a SNV are, in general, closer to the nearest SNV than comparably sized random genomic intervals. For the DMRs identified by DSS, approximately 2/3 of all DMRs contain a SNV and approximately 90% fall within 1 kb of a SNV (Fig. 3e); for DMRs identified by Metilene, more than half contain a SNV and more than 80% are within 1 kb of a genetic difference between strain (Supplementary Figure 4E). These findings illustrate the relationship between local differences in DNA sequence and allele-specific methylation in offspring, as well as support a determining role for local DNA sequence in the establishment or maintenance of local epigenetic information.

**Transcription factor–DNA interactions and DNA methylation.** As allele-specific histone modifications have been linked to genetic control of transcription factor–DNA interactions[20–24,31], we explored the possibility that a similar mechanism might correlate with the allele-specific DNA methylation observed here. We performed computational searches for known transcription factor binding sites at DMRs after extending each region of interest to a minimum size of 401 bp. A total of 20 of the approximately 140 motifs corresponding to transcription factors expressed in mouse liver were enriched over background in these genomic intervals (Fig. 4a, b; Supplementary Figure 5). The observed motifs were enriched in binding sites for transcription factors that function in liver development, hepatocyte differentiation, and liver homeostasis as opposed to transcription factors that function exclusively in other cell types. Enriched binding sites differ substantially in type of DNA-binding domain (Forkhead/winged helix; bZIP; homeobox; bHLH; Zinc finger).

We next considered what the relationship between transcription factor interaction and DNA methylation might involve. In principle, higher levels of DNA methylation at a DMR might negatively influence the capacity of a given transcription factor to bind its cognate recognition element[7–11]. Conversely, productive interaction of transcription factors with regulatory DNA could lead to local decreases in DNA methylation through physical hindrance of DNA methyltransferase[15–19] or through chromatin-based mechanisms[20–24,34]. Five enriched motifs (NRF1, Arnt:Ahr, CEBP, USF1, and USF2) have a CpG dinucleotide; binding of these transcription factors may be blocked by modification of the CpG within their consensus binding motif[7,11]. For the remaining motifs, the lack of CpG dinucleotides in the consensus binding element suggests that increased methylation within DMRs containing these motifs may result from loss of transcription factor binding.

Accordingly, we next asked whether SNVs within or adjacent to DMRs can interfere with DNA binding in a biochemical assay. Purified FoxA1 DNA-binding domain was assessed for the ability to bind to sites that fall within DMRs and have a variant sequence

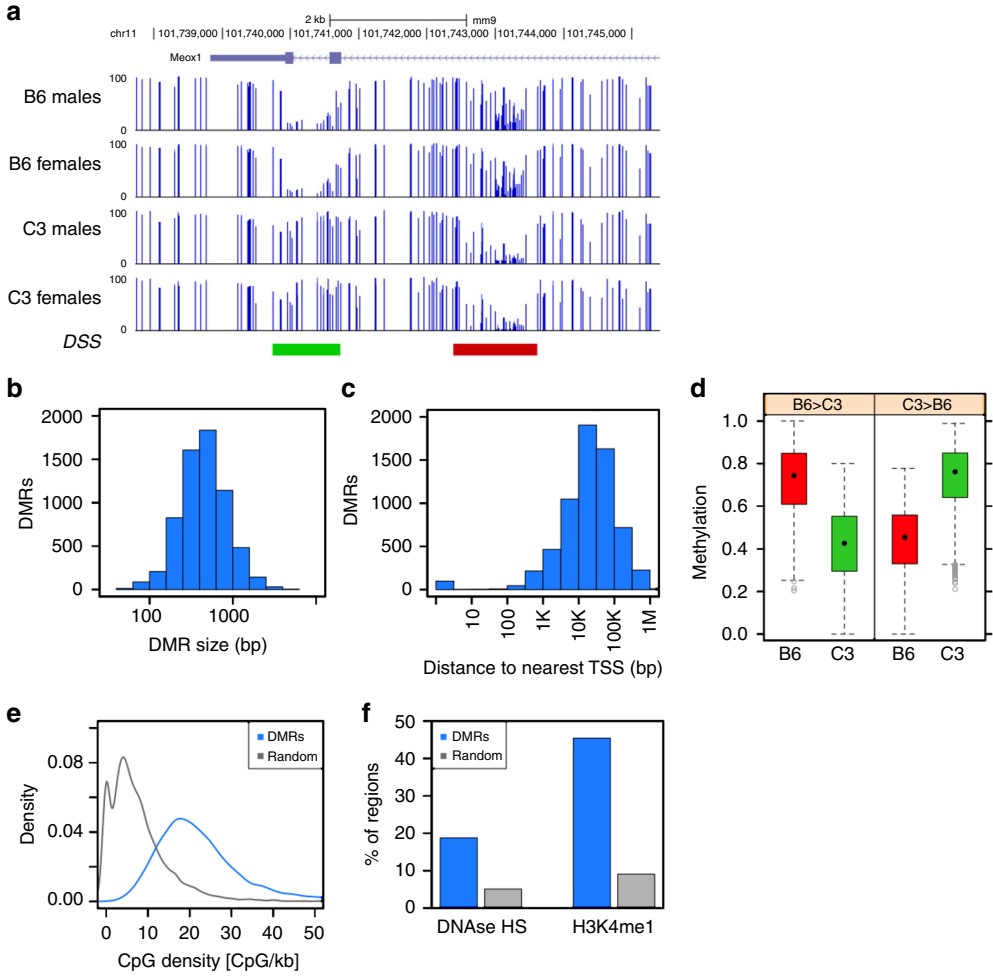

**Fig. 2** DMRs between B6 ($n = 6$) and C3 ($n = 6$) parental animals, identified by DSS. **a** Genome browser view of example B6 > C3 DMR (red) and C3 > B6 DMR (green). **b** Distribution of DMR size, $N = 6380$. **c** Distribution of distance between DMR and nearest TSS (based on RefSeq gene models as of Feb 8, 2016), $N = 6380$. **d** Distribution of methylation level in DSS DMRs, by strain and DMR polarity. $N = 3166$ B6 > C3 DMRs and 3214 C3 > B6 DMRs. The box depicts the 25th to 75th percentiles (color by genotype where B6 is red and C3 is green), the black dot is the median, the whiskers extend to data points up to 1.5*IQR beyond the box, and open gray circles are data points outside the whisker range. **e** Distribution of CpG density at DSS DMRs ($N = 6380$), compared to size-matched random genomic regions ($N = 6380$). **f** Overlap of DSS DMRs ($N = 6380$) with DNase hypersensitive sites (UCSC Genome Browser) and H3K4me1 enriched regions (ENCODE peaks ENCFF001XXZ), compared to the average overlap observed for 1000 iterations of size-matched random genomic regions

between the two strains within the binding site (Fig. 4c, e, f, Supplementary Figure 6). In most cases, recombinant FoxA1 DNA-binding domain exhibited a preference for the sequence found in the strain with lower methylation levels at the DMR tested. We asked whether these transcription factor binding results were reflected in vivo by performing ChIP-seq for FoxA1 in liver from B6 and C3 male animals. Analysis of binding at FoxA1 sites within or near a DMR that contain a SNV revealed that FoxA1 binding to these sites was impacted by the genetic differences between strain in the same manner as in the in vitro biochemical assays (Fig. 4d). Where the allele was more methylated in C3 animals, ChIP signal was higher in B6 animals and vice versa (Fig. 4e, f). At these loci, the ChIP-seq data are entirely consistent with genetic control of FoxA1 binding within/ near DMRs correlating inversely with local methylation status. These data suggest that local DNA sequence within a select group of genomic transcription factor binding sites has an influence on local DNA methylation. This finding is consistent with a multitude of literature reports linking local DNA methylation state to transcription factor binding[16–19,23].

**Differentially methylated regions by sex and life experience.** We capitalized on the design of our breeding experiment to ask whether autosomal DNA methylation patterns differ by sex. As pregnancy is known to impact liver biology in rodents[35], we first assessed whether the livers of dams studied here had normalized from pregnancy-associated hypertrophy by histologic examination of sections prepared from the right liver lobe of all study animals (see Methods). We observed no differences in histologic features in any group suggesting general features of cellular content are similar in all study animals. When comparing dams to virgin females or to males, we observed no difference in cell size, suggesting that hepatocytes in dams had returned to baseline at the cellular level at the time of sacrifice (Supplementary Figure 7). Accordingly, we utilized both DSS and Metilene to identify DMCs and differentially methylated regions (DMRs) upon comparison of the dams in our breeding experiment (both the C3 and B6 female animals) with the group of sires (both C3 and B6 male animals). DSS identified 12,224 DMCs based on sex which underlie 1575 DMRs; Metilene identified 51, 675 DMCs and 439 DMRs (Table 1). The total number of DMRs, regardless of

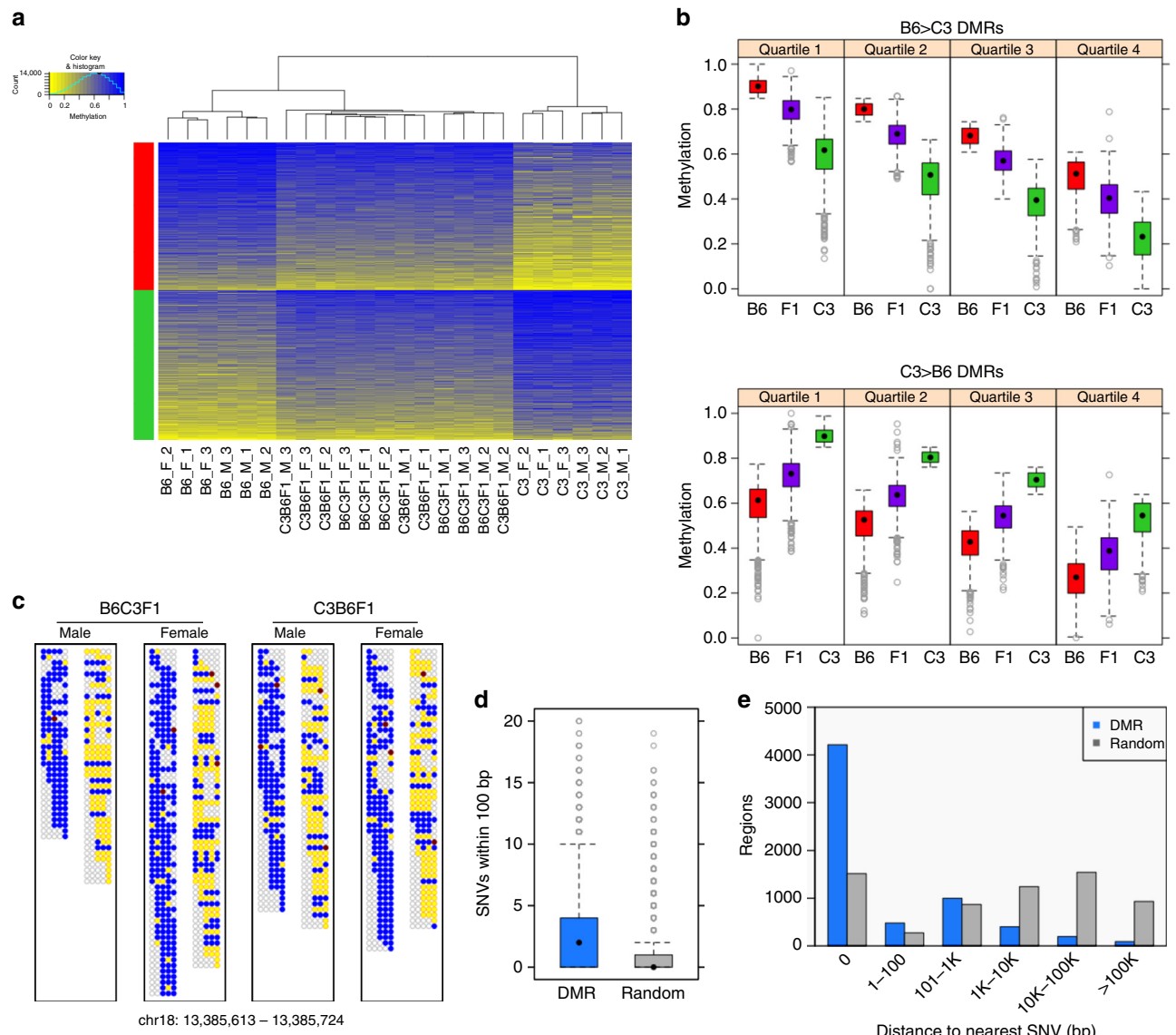

**Fig. 3** DNA methylation patterns at DSS genotype–DMRs are recapitulated on parental alleles in F1 progeny. **a** Heatmap view of weighted methylation scores per B6-vs.-C3 DSS DMR ($N = 6380$) per animal. Hierarchical clustering of animals performed by R package "amap" (hcluster with method = euclidean and link = average). DMRs were split by direction (B6 > C3: red bar, or C3 > B6: green bar) then sorted by average methylation score over all 24 animals. **b** Distribution of average weighted methylation scores, with DMRs ($N = 6380$) split into quartiles according to decreasing methylation score of the hypermethylated parental genome. The comparison between B6 and F1 or between C3 and F1 are significant at $p < 1e{-}80$ (Mann–Whitney) in all quartiles. **c** Methylation in F1 progeny at the read level for an exemplar DSS DMR (B6 > C3). Each box represents the collection of read fragments (for a given F1 genotype) that could be unambiguously assigned as originating from either the B6 parent (left side) or the C3 parent (right side) based on the presence of a diagnostic SNV. Each row within a box and column represents one sequenced fragment, with each CpG site indicated by a circle colored according to the following scheme: yellow = unmethylated cytosine, blue = methylated cytosine, red = noncytosine, and empty/gray = no mapped base at given CpG site in given fragment. **d** Number of SNVs local to DMRs ($N = 6380$), compared to size-matched random genomic regions ($N = 6380$). $p < 1e{-}300$ (Mann–Whitney test). **e** Distribution of distance to the nearest SNV for DMRs ($N = 6380$) or size-matched random genomic regions ($N = 6380$). Distance = 0 indicates that at least one SNV is found within a region of a given type. In the box-and-whisker plots, the box depicts the 25th to 75th percentiles, the black dot is the median, the whiskers extend to data points up to 1.5*IQR beyond the box, and open gray circles are data points outside the whisker range

analysis stream, were asymmetric in terms of polarity with female > male DMRs being found roughly twice as often as the converse (Supplementary Data 3). This dataset confirms that our animals have larger differences in autosomal DNA methylation pattern based on genotype than on sex in the tissue studied (i.e., 1575/439 DMRs based on sex vs. 6380/2569 DMRs based on genotype).

Exemplar DMRs are illustrated in browser format for each analysis stream (Fig. 5a, Supplementary Figure 8A). Like regions that differ in comparison across genotype, sex-dependent DMRs

are relatively small, the overwhelming majority under 1 kb in size (Fig. 5b, Supplementary Figure 8B). Sex-dependent DMRs, like genotype-dependent DMRs, fall distant from mapped transcription start sites with greater than 60% falling more than 10 kb from the nearest TSS (Fig. 5c; Supplementary Figure 8C). Differences in methylation level between the two groups compared is similar in sex–DMRs and in genotype–DMRs (Fig. 5d; Supplementary Figure 8D). Like their genotype-driven counterparts, sex–DMRs have higher CpG density than random controls but lack the

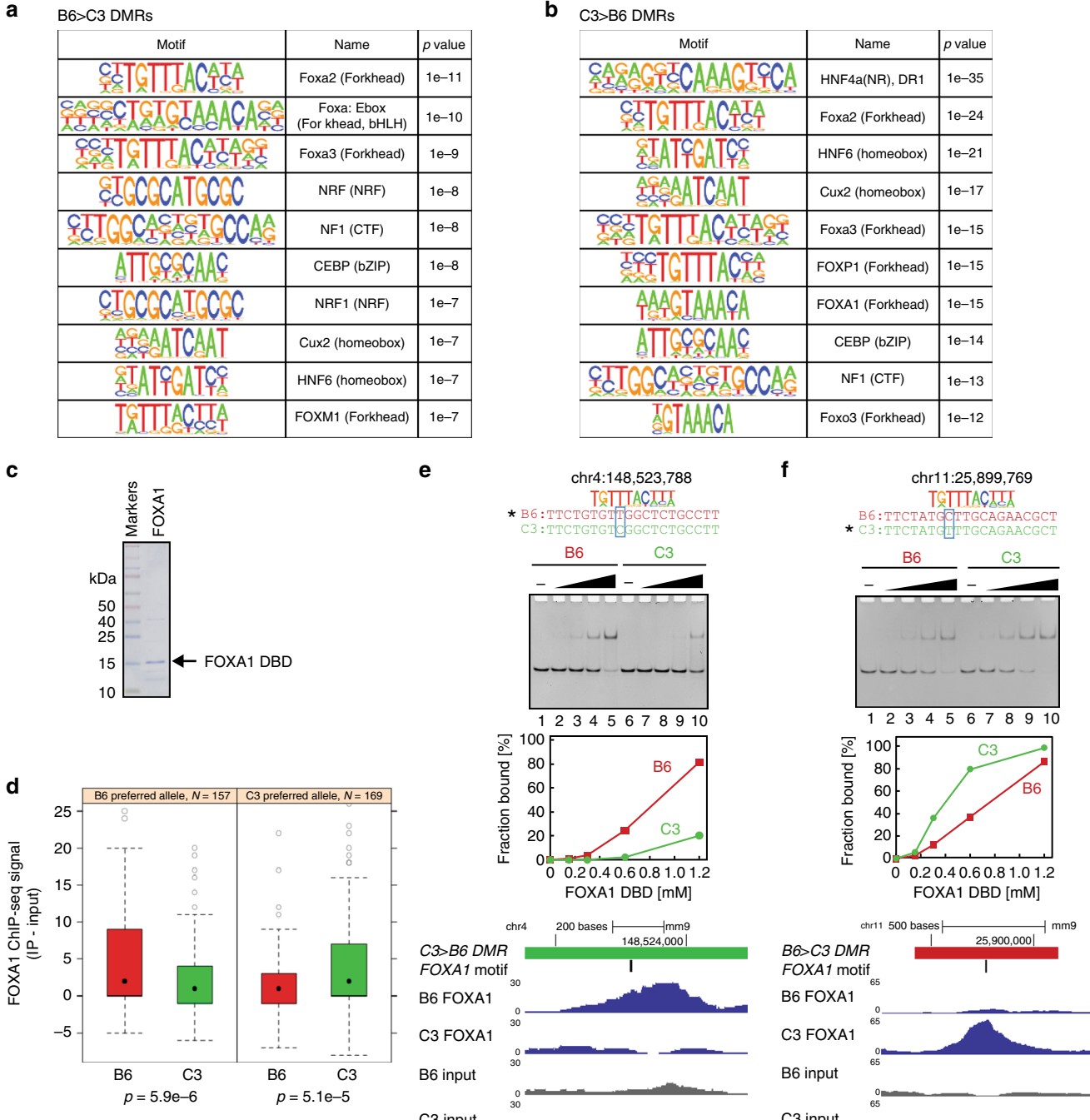

**Fig. 4** SNVs at DMRs affect DNA-binding activity of a tissue-specific transcription factor. **a** Top 10 enriched TF motifs in B6 > C3 DSS DMRs (after expanding DMR size to a minimum of 401 bp) according to HOMER v4.9.1. **b** Top 10 enriched TF motifs in C3 > B6 DSS DMRs (after expanding DMR size to a minimum of 401 bp) according to HOMER v4.9.1 **c** Protein purification of FOXA1 DNA-binding domain (DBD). Purified FOXA1 DBD was analyzed by SDS-PAGE. The black arrow indicates FOXA1 DBD. **d** FOXA1 ChIP-seq signal at SNV-containing FOXA1 motifs. Occurrences of FOXA1 motifs (as defined by HOMER motifs FOXA1.LNCAP or FOXA1.MCF7) within 100 nt of a DMR were categorized by whether the B6 or C3 allele was preferred according to the HOMER position weight matrix [PWM]. FOXA1 signal in the B6 and C3 animals is reported as input-subtracted ChIP-seq read count at 100 nt regions centered on the motif occurrences. The box depicts the 25th to 75th percentiles, the black dot is the median, the whiskers extend to data points up to 1.5*IQR beyond the box, and open gray circles are data points outside the whisker range. Reported p values are from Mann–Whitney test. **e**, **f** DNA-binding analyses of FOXA1 DBD with SNV-containing DNAs. The genomic positions and DNA substrates are indicated at the top of each panel with a representative FOXA1 motif. The position of the SNV is highlighted with light blue boxes. The band intensities were calculated using ImageJ software. Totally, 20 bp double-stranded DNAs were incubated with the FOXA1 DBD. The concentration of FOXA1 DBD was as follows: 0 μM, lanes 1,6; 0.15 μM lanes 2,7; 0.3 μM, lanes 3,8; 0.6 μM, lanes 4,9; 1.2 μM, lanes 5,10. The protein–DNA complex was separated by native-polyacrylamide gel electrophoresis. Browser tracks of ChIP-seq data for each locus is shown below the DNA-binding data

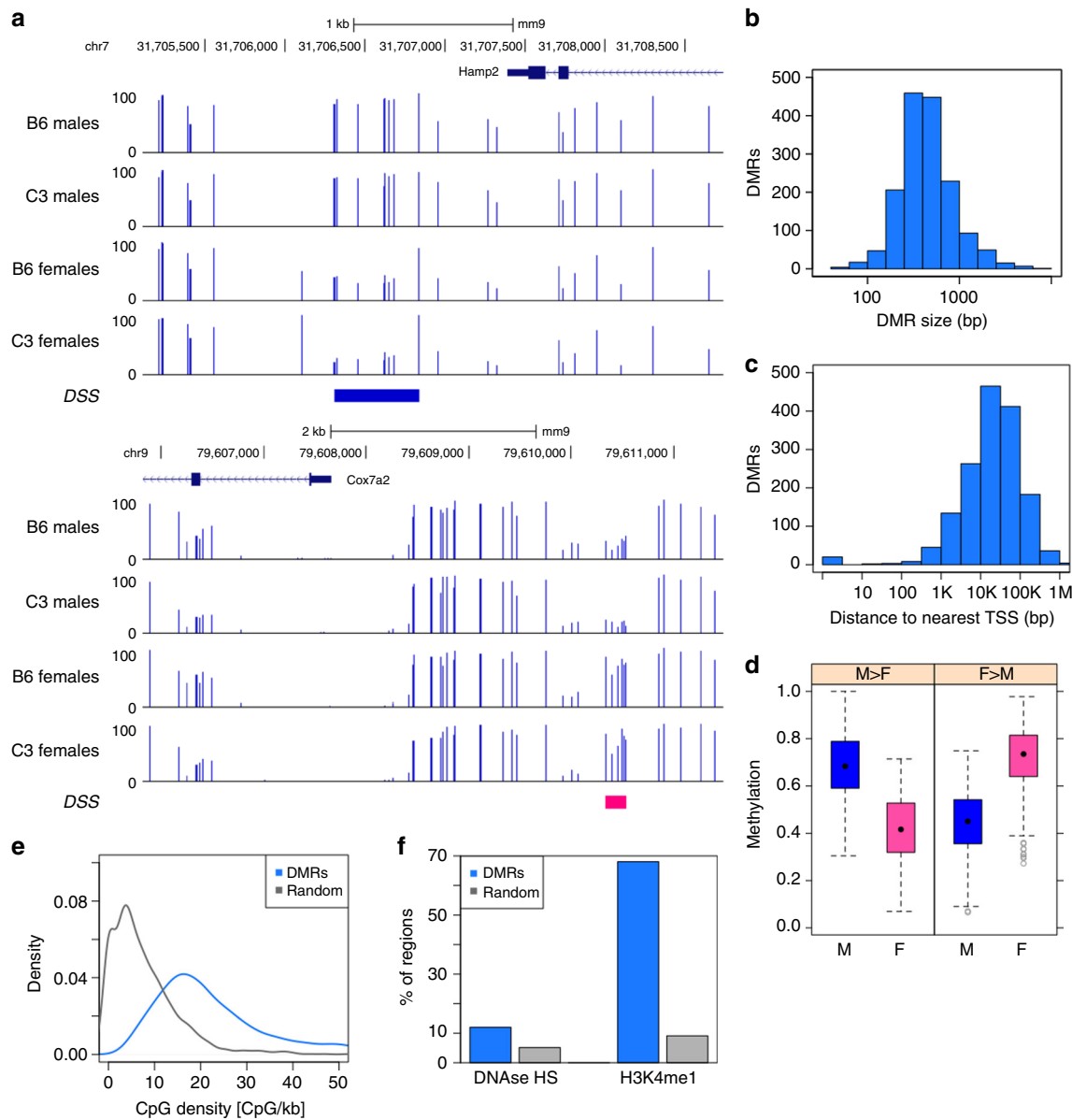

**Fig. 5** DMRs between male ($n = 6$) and female ($n = 6$) parental animals, identified by DSS. **a** Genome browser view of example M > F DMR (blue, top) and F > M DMR (pink, bottom). **b** Distribution of DMR size, $N = 1575$. **c** Distribution of distance between DMR and nearest TSS (based on RefSeq gene models as of Feb 8, 2016), $N = 1575$. **d** Distribution of methylation level in DSS DMRs, by strain and DMR polarity. $N = 1041$ F > M DMRs and 534 M > F DMRs. The box depicts the 25th to 75th percentiles (color by sex where male is blue and female is pink), the black dot is the median, the whiskers extend to data points up to 1.5*IQR beyond the box, and open gray circles are data points outside the whisker range. **e** Distribution of CpG density at DSS DMRs ($N = 1575$), compared to size-matched random genomic regions ($N = 1575$). **f** Overlap of DSS DMRs ($N = 1575$) with DNase hypersensitive sites (UCSC Genome Browser) and H3K4me1 enriched regions (ENCODE peaks ENCFF001XXZ), compared to the average overlap observed for 1000 iterations of size-matched random genomic regions

density of CpG islands (Fig. 5e; Supplementary Figure 8E). Finally, like genotype–DMRs, sex–DMRs also colocalize to an appreciable extent with DNAse hypersensitivity and with the enhancer mark, H3K4me1 (Fig. 5f; Supplementary Figure 8F).

We next performed clustering and visualization of DNA methylation levels of all 24 animals in the study at the sex–DMRs precisely as we did with genotype–DMRs (Fig. 6a; Supplementary Figure 9A). We observed that female > male DMRs behaved as expected; all female animals had high methylation levels while all male animals had lower levels (Fig. 6a, b; Supplementary Figure 9A, B). Surprisingly, the male > female DMRs had a different outcome. We observed, at this subset of loci, that female hybrid F1 animals had methylation levels that more closely

resembled males than the parental female animals (Fig. 6a, b; Supplementary Figure 9A, B). Unlike the genotype–DMRs described above, there was no apparent allele specificity to DNA methylation at sex–DMRs (Fig. 6c; Supplementary Figure 9C), nor was there a proximity relationship to genetic differences between strain (Fig. 6d, e; Supplementary Figure 9D, E).

To investigate the unexpected behavior of sex–DMRs in female animals, we used the Genomic Regions Enrichment of Annotations Tool[36] to predict functions of the male > female sex–DMRs. Genes predicted to be regulated by these loci were enriched in Gene Ontology terms related to pregnancy and lactation (Fig. 7a). An exemplar gene from this category is the prolactin receptor gene located on chromosome 15 (Fig. 7b) which contains

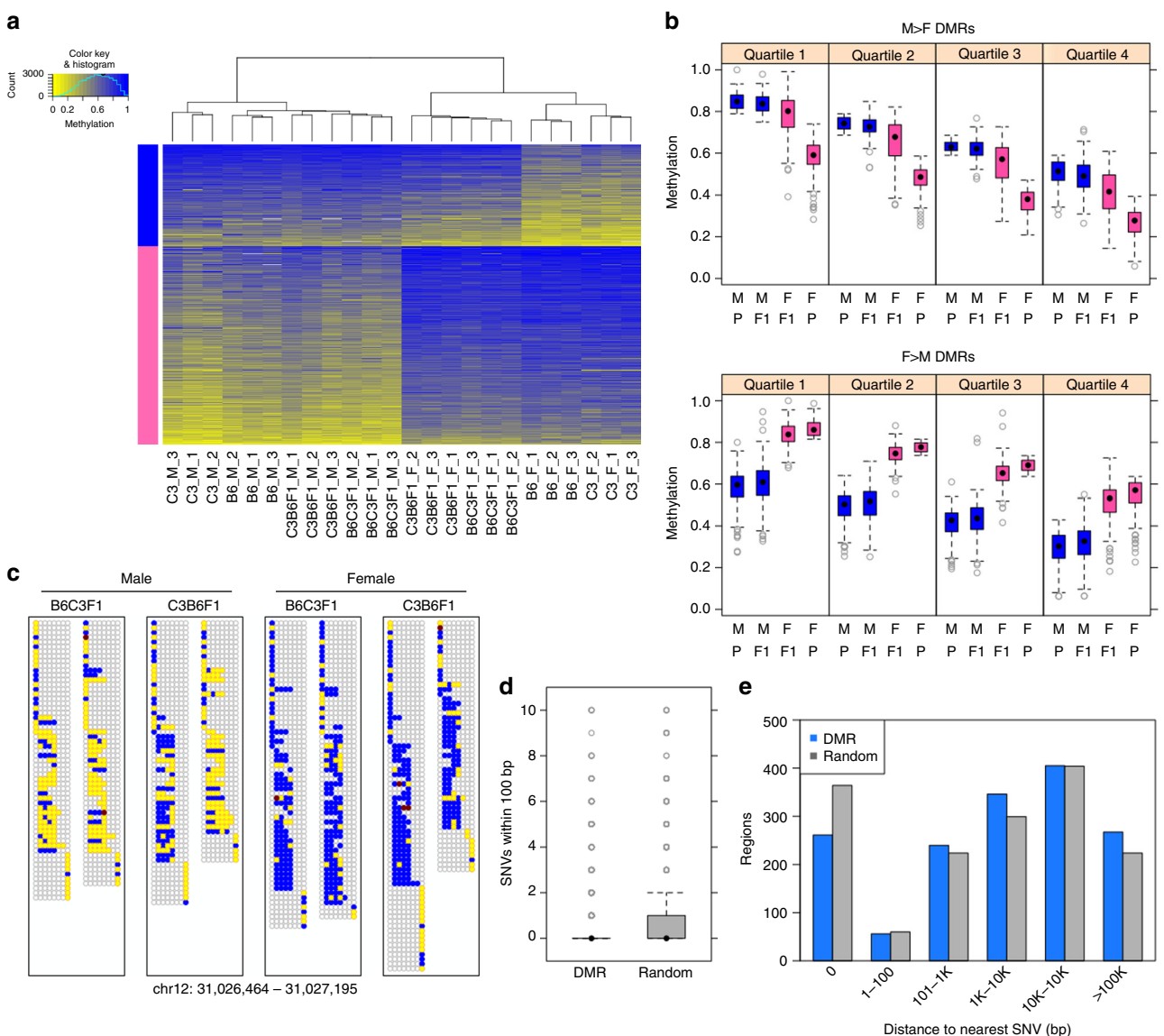

**Fig. 6** Recapitulation of DNA methylation patterns at DSS sex-DMRs in F1 progeny varies by DMR polarity. **a** Heatmap view of weighted methylation scores per male-vs.-female DSS DMR ($N = 1575$) per animal. Hierarchical clustering of animals performed by R package "amap" (hcluster with method = euclidean and link = average). DMRs were split by direction (M > F: blue bar, or F > M: pink bar) then sorted by average methylation score over all 24 animals. **b** Distribution of average weighted methylation scores, with DMRs split into quartiles according to decreasing methylation score of the hypermethylated parental sex. Animals are grouped according to both sex (M, F) and genotype group (P = parental B6 or C3, F1 = offspring B6C3F1 or C3B6F1). **c** Methylation in F1 progeny at the read level for an exemplar DSS DMR, organized as described for Fig3C. **d** Number of SNVs local to DMRs ($N = 1575$), compared to size-matched random genomic regions ($N = 1575$). **e** Distribution of distance to the nearest SNV for DMRs ($N = 1575$) or size-matched random genomic regions ($N = 1575$). Distance = 0 indicates that at least one SNV is found within a region of a given type. In the box-and-whisker plots, the box depicts the 25th to 75th percentiles, the black dot is the median, the whiskers extend to data points up to 1.5*IQR beyond the box, and open gray circles are data points outside the whisker range

multiple male > female DMRs called by both analysis streams. Prolactin is known be upregulated in liver in pregnant and lactating rodents[37]. This finding is consistent with the design of our study wherein the female B6 and C3 animals analyzed were the dams of the female F1 animals (which were virgin females). These findings suggest that the class of male > female DMRs are dominated by genomic intervals in liver that lose DNA methylation downstream of pregnancy, lactation, or both.

We further explored the relationship of DNA methylation with pregnancy and lactation by defining DMRs in the comparison of dams (again including B6 and C3 animals) with daughters (F1 hybrid virgin females sacrificed at the same age). DSS identified 2257 DMCs and 305 DMRs in this comparison; Metilene

identified 17,798 DMCs and 68 DMRs (Table 1). As anticipated, the polarity of DMRs was non-random, with 295 of 305 (and 66 of 68) DMRs having greater CpG methylation in virgin females than in dams (Supplementary Data 4). Of the 305 DMRs called by DSS in this comparison of dams versus F1 hybrid virgin females, 196 overlap M > F DMRs and 0 overlap F > M DMRs. This analysis suggests that DNA methylation patterns, when queried in an appropriate manner, can provide a record of life events such as pregnancy/lactation. This record appears to correlate with loci involved in the relevant biological response to pregnancy/lactation.

The sex–DMRs were also analyzed for enrichment of transcription factor motifs. The overwhelming majority of motifs

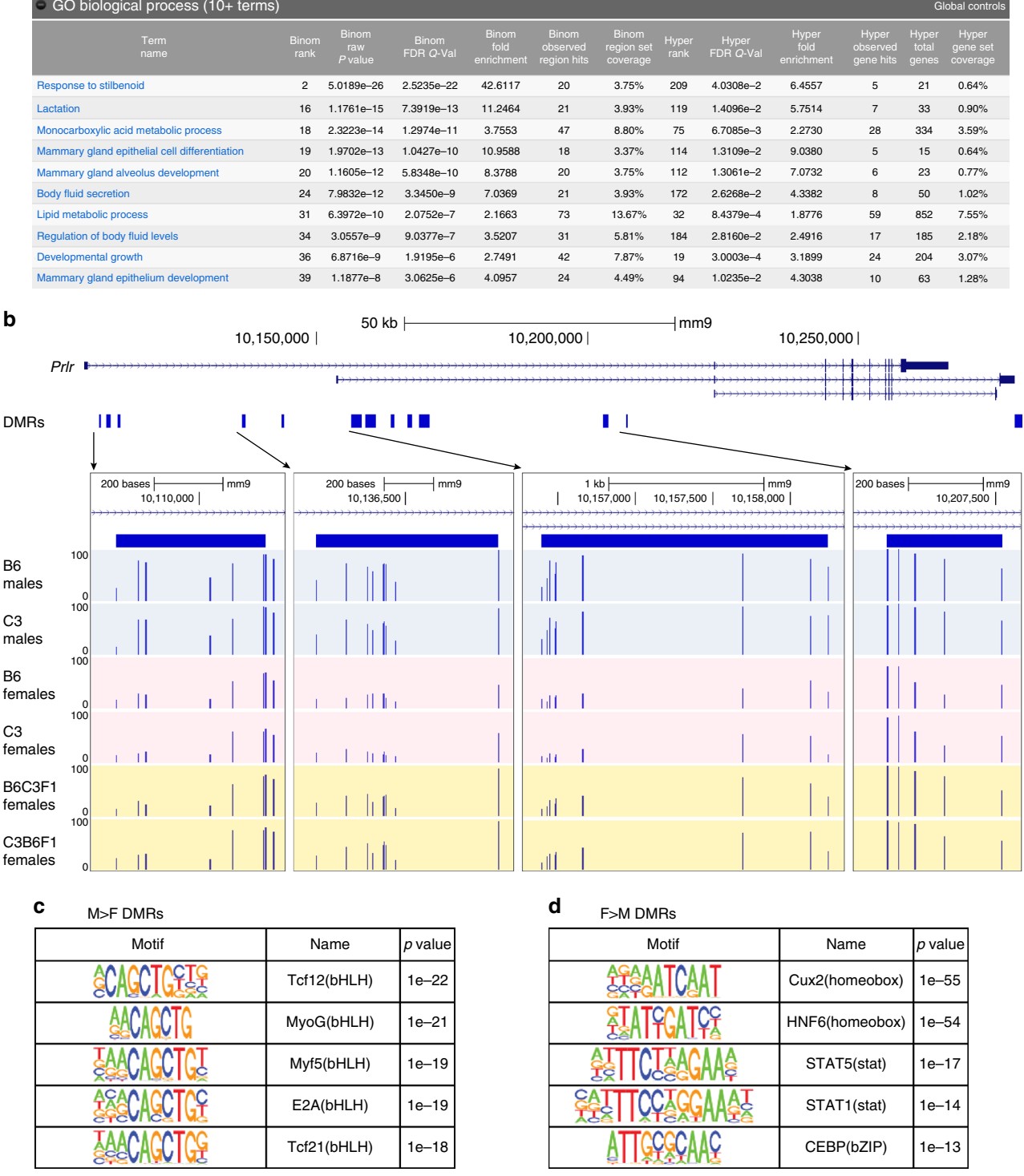

**Fig. 7** DMRs hypomethylated in females suggest link to pregnancy and lactation. **a** Top ten biological processed reported by GREAT v3.0.0 for DSS male > female DMRs. Bold text indicates significance scores. **b** Genome browser view of prolactin receptor gene locus, with zoomed panels showing CpG methylation level in parental males, parental females, and F1 females at exemplar male > female DMRs. **c** Top five enriched TF motifs in male > female DSS DMRs (after expanding DMR size to a minimum of 401 bp) according to HOMER v4.9.1. **d** Top five enriched TF motifs in female > male Metilene DSS (after expanding DMR size to a minimum of 401 bp) according to HOMER v4.9.1

identified in the male > female DMRs contain some version of the E-box sequence—CANNTG (Fig. 7c; Supplementary Figure 10A). Motifs in the female > male sex–DMRs are much more diverse, containing binding sites for homeobox factors, nuclear receptors, signal responsive transcription factors, bHLH, and bZIP factors

(Fig. 7d; Supplementary Figure 10B). While many motifs enriched in this analysis do not contain CpG within their consensus motif, we note that Hox transcription factors can bind to methylated DNA, despite the lack of CG dinucleotides in their consensus recognition site[11].

## Discussion

The results presented here demonstrate that epigenetic features such as DNA methylation are closely linked to DNA sequence with patterns in parents recapitulated on the relevant allele in offspring. Our finding that DMRs colocalize to a large extent with distal regulatory DNA and transcription factor binding motifs suggests a potentially causal relationship between DNA sequence-dependent alterations in the interactions of DNA-binding proteins with their respective target sequences and changes in local DNA methylation[2,23,24,31]. Further, the data suggest that differential CpG methylation, acting downstream of genetic changes in transcription factor–DNA interaction, demarcate enhancers that are active in only one strain, or on a single allele in F1 progeny.

The differences in DNA methylation pattern between inbred strains, occurring largely at enhancer loci where DNA methylation pattern may be programmed by transcription factor action, resemble the differences between cell/tissue type within strain which also closely colocalize with enhancers[31,32]. Our data add to the growing body of evidence associating alterations in the quality of interaction of a transcription factor with its DNA target with changes in epigenetic features, including DNA methylation[20–24,31]. We observe evidence for differences in local DNA methylation correlating with binding quality for only a subset of transcription factors expressed in mouse liver. Thus, our findings refine available models, and suggest that not all transcription factors have the inherent ability to reprogram local epigenetic state.

The linkage of CpG methylation to local DNA sequence in somatic tissues hinges, in part, on the capacity of genetic information to influence DNA/protein interactions and recruitment of chromatin modifiers either in the tissue of interest or in a developmental precursor undergoing programmed refinement of the pattern of DNA methylation. This model has profound implications for interpretation of changing chromatin marks, including DNA methylation, in studies of animals derived from outbred populations, including humans. Our results further demonstrate that patterns of DNA methylation are remarkably similar across generation. We find that the overwhelming majority of CpG methylation is preserved in comparison of parent with offspring. However, we also find that appropriate comparisons can reveal DNA methylation changes that correlate with a major life event even weeks afterwards, in this case pregnancy and/or lactation. Of interest, the DNA methylation changes distinguishing dams from their virgin daughters were persistent when assessed 4 weeks after weaning and after cytologic alterations in hepatocytes characteristic of pregnancy had returned to baseline. While we have not identified, to date, the mechanistic basis behind focal loss of CpG methylation at a few hundred loci in female animals that have borne and nurtured offspring, it is tempting to speculate action of a transcription factor(s) acting to regulate expression level of genes integral to that biology.

The data and analyses presented here speak to the stability of DNA methylation patterns across generation within a somatic tissue. CpG methylation was found to track with DNA sequence, local differences in epigenetic marks correlate with genetic control of transcription factor/DNA interactions. The pattern of DNA methylation was stably associated with DNA sequence whether it passed through the male or female germ line. In contrast, we observed that life events can have a profound influence on DNA methylation in somatic cells, unlike the case of germ cells[38,39].

## Methods

**Animal care.** C57BL/6N and C3H/HeN mice were obtained from the NTP colony at Taconic Farms, Inc. (Germantown, NY). Animals were housed in an AALAC accredited facility at Integrated Laboratory Systems (Durham, NC; Project-Study#N135-234), all procedures were in compliance with the Animal Welfare Act Regulations, 9 CFR 1–4 with handling and treatment according to the *Guide for the Care and Use of Laboratory Animals*[40], and were approved by the Animal Care and Use Committee, Integrated Laboratory Systems. Animals were maintained in climate controlled rooms (18–26 °C; 35–65% humidity) on a 12:12 h light:dark cycle in polycarbonate cages with irradiated, heat-treated hardwood bedding (Teklad, Indianapolis, IN). Cotton fiber nestlets (Ancare Corp., Bellmore, NY) were supplied to mice for environmental enrichment. NIH 31 (Harlan Teklad, Indianapolis, IN) was provided ad libitum to all breeding pairs and to their B6C3F1/N or C3B6F1/N progeny from birth until 17 weeks of age. Beginning at 18 weeks of age, all animals were provided NTP 2000 (Zeigler Bros., Gardners, PA) ad libitum in order to simulate dietary conditions of NTP breeding and B6C3F1/N and C3B6F1 hybrid mice assigned to NTP studies. Reverse osmosis treated water from sterilized bottles with stainless steel sipper tubes was provided ad libitum to all animals and changed weekly.

For breeding, five pairs of female C57BL/6N and male C3H/HeN mice at week 10 were outcrossed to produce B6C3F1 mice. The reverse outcross was performed independently using five pairs of female C3H/HeN and male C57BL/6N mice at week 10 and C3B6F1 mice were collected. Animals were multihoused up to two adults per cage during pair mating and single housed when not mating. B6C3F1 or C3B6F1 offspring were multi-housed up to four per sex per cage. Each mouse was uniquely identified by ear tag prior to the start of the study. The F1 progeny were associated with their dam and sire and tracked.

Animals for in-depth analysis were selected based on availability of offspring of both sexes from each cross performed. We selected three nuclear families (sire, dam, female F1, and male F1) to avoid the possibility that a biological outlier animal would significantly impact the results. Of available families, we randomly selected three from each cross. Animals were mated at age of 10 weeks. Weaning occurred at week 16. All animals in the study were sacrificed at week 20. Investigators were not blinded to family selection.

**Histology.** Three representative sections were cut at 5 μm$^2$ thickness from formalin-fixed, paraffin-embedded blocks. Section 1 was stained with hematoxylin and eosin (H&E). Female mice from three groups of animals: virgins (B6 and C3), breeders (B6 dams and C3 dams) and first generation (B6C3F1 and C3B6F1), were evaluated for hyperplasia and hypertrophy. For assessment of hyperplasia, liver sections were examined using a Leica-DMLB microscope with a 40× objective. Ten fields per slide were analyzed for hepatocyte mitotic figures. Hypertrophy assessment was based on hepatocyte density counts generated from the same, H&E-stained liver sections scanned on the Aperio ScanScope XT instrument (Vista, CA) using ImageScope software, (v11.2.0.780, Aperio). Hepatocyte nuclei from 10, nonoverlapping, 90 × 90 μm$^2$ fields were analyzed per section to determine cell density.

**Genomic DNA extraction.** Frozen liver tissues were quickly pulverized with BioPulverizer (Bio Spec Products Inc.) on dry ice. DNA was extracted with Allprep DNA/RNA/Protein Mini Kit (Qiagen) according to the manufacturer's protocol. DNA was further purified by phenol–chloroform extraction followed by ethanol precipitation.

**Whole-genome bisulfite sequencing library preparation.** DNA (1 μg) was spiked with 1 ng unmethylated lamda DNA (Promega), fragmented (average size; 300 bp), end-repaired, A-tailed, and adapter-ligated using Truseq DNA sample prep kit (Illumina) according to the manufacturer's protocol. Adapter-ligated DNA was gel isolated, (2% agarose gel, DNA ranged from 400 to 500 bp) and recovered using a QIAquick gel extraction kit (Qiagen). After clean up with AMPure XP beads, bisulfite conversion was performed using EpiTect Bisulfite kit (Qiagen) with the following thermal cycles, 95 °C 5 min, 60 °C 25 min, 95 °C 5 min, 60 °C 85 min, 95 °C 5 min, 60 °C 175 min, 95 °C 5 min, 60 °C 180 min. After clean up with AMPure XP beads, bisulfite converted DNA was amplified with PfuTurbo Cx Hotstart DNA Polymerase with following thermal cycles, 95 °C 5 min, 98 °C 30 s, 12 cycles of (98 °C 10 s, 65 °C 30 s, 72 °C 30 s), 72 °C 5 min. DNA was cleaned with AMpure XP beads, and stored at −30 °C until use. Sequencing was performed in HiSeq2000 using PE100 base format.

**Whole-genome bisulfite sequencing data processing.** General quality control checks were performed with FastQC v0.8.0 (http://www.bioinformatics.babraham.ac.uk/projects/fastqc/). Each dataset was filtered for average base quality score (>20). Filtered datasets were aligned to a reference genome using Bismark v0.7.8 (parameters -X 10000 --non_bs_mm -n 2 -l 50 -e 70 --chunkmbs 1024)[41], using Bowtie v0.12.8[42] as the underlying alignment tool. The reference genome index contained the genome sequence of enterobacteria phage λ (NC_001416.1) in addition to all chromosomes of the mm9 assembly (NCBI 37). Mappings for all datasets generated from the same library were merged, and duplicates removed via the Bismark deduplication tool (deduplicate_bismark_alignment_output.pl). Mapped reads were then separated by genome (mm9 or phage λ) and by source strand (plus or minus). The first four and last one base of each read2 in all read pairs was clipped due to positional methylation bias, and any redundant mapped bases due to overlapping mates from the same read pair were trimmed to avoid bias in quantification of methylation status. Finally, the

SAM alignments for multiple libraries from the same animal were merged. Read pairs mapped to phage λ were used as a QC assessment to confirm that the observed bisulfite conversion rate was >99%. Read pairs mapped to the mm9 reference genome were used for downstream analysis.

**Validation of genomic cytosine context**. For each cytosine site in the mm9 reference genome, the fractions of mapped bases that are C at position N, G at position N + 1, and G at position N + 2 were calculated using WGBS data from all six B6 or from all six C3 parent animals. The cytosine context for a given genomic position is considered to be validated if the following criteria are met: (a) the expected cytosine context is consistent between the mm9 reference genome, the local C57BL/6 N assembly, and local C3H/HeN assembly; (b) at least 75% of mapped bases at each of the N, N + 1, and N + 2 positions are consistent with the expected context using the B6 WGBS data; and (c) at least 75% of mapped bases at each of the N, N + 1, and N + 2 positions are consistent with the expected context using the C3 WGBS data. For CpG context, validation was required on only one of the two strands. For CHG and CHH context, each strand was evaluated independently. This process identified 18,887,127 validated autosomal CpG sites, 101,235,798 and 101,246,101 validated autosomal CHG sites (on the plus and minus strand, respectively), and 351,689,190 and 351,716,331 validated autosomal CHH sites (on the plus and minus strand, respectively). In this study, methylation analysis was limited to only autosomal cytosines with validated CpG context.

**DMR detection**. Using the DSS R package v2.15.0, DMCs were identified by DSS with the callDML function (default parameters), and DMRs were identified with the callDMR function (pct.sig = 0.75, all other parameters default). DMC and DMR calls were also made via Metilene v.0.2-6 (-m 5 for DMR calls, all other parameters default), with a $p$ value threshold of 0.01 and mean methylation difference of 0.2 for DMCs and a $q$ value of 0.05 for DMRs. All DMR calls from both tools were subject to additional filters, as described below. DMRs were required to contain at least five validated CpG sites and have a weighted methylation score[33] difference of at least 20%. Furthermore, DMRs were excluded if there was an extreme sequencing depth imbalance between sample groups: specifically, if either sample group had average depth > 500 (which is more than 3× the genomic average), or if there was a >5-fold differences in average depth between sample groups.

**Enriched motifs**. Each set of DMR calls was split by polarity (i.e., B6 > C3 or C3 > B6, male > female or female > male). Each DMR was extended to a minimum width of 401 bp, centered on the midpoint of the called DMR. Within each of DMRs, any regions that were overlapping after this extension were merged. HOMER[43] v4.9.1 (parameters -size given -gc -nomotif) was run to identify enriched motifs among HOMER's library of known motifs.

**Recombinant protein purification of FOXA1 DBD**. Human FOXA1 gene fragment encoding amino acids 170–270 was inserted into the pDEST17 expression vector to produce a hexahistidine tagged fusion protein. The protein was expressed in the *Escherichia coli* BL21 (DE3) Codon-Plus RIL cells. The cells were lysed in a high salt (1 M NaCl) buffer containing 10 mM imidazole. The His-tagged proteins were purified by Ni-NTA agarose column chromatography (Qiagen). The eluted proteins were further purified by HiTrap Heparin HP column chromatography (GE Healthcare Life Sciences). The eluted proteins were snap-frozen in liquid nitrogen and stored at −80 °C.

**DNA-binding assay**. Double-stranded (ds) DNAs were prepared by annealing complementary oligonucleotides in a 1:1 molar ratio. DsDNAs (0.3 μM) were incubated with the indicated concentration of FOXA1 DBD in a reaction buffer containing 20 mM Tris-HCl (pH 7.5), 4 mM potassium phosphate (pH 7.2), 7% glycerol, 1 mM DTT, 0.4 mM 2-mercaptoethanol, 1 mM MgCl2, 0.1 mg/ml BSA, and 200 mM NaCl. After 20 min incubation at room temperature, the samples were analyzed by 12% native-polyacrylamide gel electrophoresis in 0.5× TBE buffer.

**ChIP-seq**. Frozen livers were thawed on ice and homogenized in PBS. The cell debris was removed by centrifugation and the cells were fixed with 1% formaldehyde at room temperature for 10 min. Glycine was added to quench the reaction. The fixed cells were further treated with hypotonic buffer (10 mM HEPES-NaOH pH 7.9, 10 mM KCl, 1.5 mM MgCl2, 340 mM sucrose, 10% glycerol, 0.5% Triton X-100, and protease inhibitor cocktail). Nuclear pellets were resuspended in lysis buffer containing 0.1% SDS and sonicated by Covaris S220. Immunoprecipitation was performed with 2.5 micrograms anti-FOXA1 antibody (abcam, ab5089) per IP. The sequencing libraries were prepared by the NEXTflex Rapid DNA-seq kit (Bioo Scientific Corporation) and sequenced on NovaSeq 6000 (Illumina) at the NIEHS Epigenomics Core Facility. General quality control checks were performed with FastQC. Adapter was removed from raw sequencing data with cutadapt v1.2.1 (https://doi.org/10.14806/ej.17.1.200, parameters -a GATCGGAAGAG -O 5 q 0). The trimmed reads were further clipped to ensure there were no dovetailed read pairs resulting from inconsistent adapter trimming, and an average base quality score (>20) minimum filter was applied. Trimmed and filtered read pairs were aligned to the mm9 reference assembly with Bowtie v1.2[42] (parameters -m 1 -X 1000 --chunkmbs 1024). The Picard tool suite v1.110 (http://broadinstitute.github.io/picard) was used to remove duplicate read pairs (MarkDuplicates.jar) and subsequently merge replicate samples (MergeSamFiles.jar). For visualization and quantification purposes, each read pair was converted to a single fragment then each dataset was randomly downsampled to match the sample condition with the lowest read counts (N = 33,055,298).

**Reporting Summary**. Further information on experimental design is available in the Nature Research Reporting Summary linked to this Article.

## Data availability
The genomic data discussed in this publication have been deposited in the Gene Expression Omnibus (Accession Number GSE106379). Source data are provided as a Source Data file. A Reporting Summary for this Article is available as a Supplementary Information file. All other data are available from the corresponding author upon reasonable request.

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

## Acknowledgments

This work was supported, in part, by the Intramural Research Program of the National Institute of Environmental Health Sciences, NIH (ES103146 to P.A.W.). The authors express gratitude to Drs. Richard Woychik, Trevor Archer, Angela Ting, and Paula Vertino for useful comments throughout the course of this work.

## Author contributions

Study design: S.A., J.R.B., J.E.F., B.A.M., R.R.T., J.W.T., J.C.M., and P.A.W. Data generation: T.S., M.T., C.G.D., and N.I.S.C. comparative sequencing program. Data analysis: S.A.G., J.W.T., B.B., J.R.B., A.B., Y.D., C.G.D., J.F.F., J.L., B.A.M., R.R.T., X.X., P.B., D.C.F., J.C.M., and P.A.W. Data visualization: S.A.G., F.D., Y.D., T.W., and X.X. Manuscript preparation: All authors participated in drafting and editing of the manuscript.

## Additional information

**Competing interests:** The authors declare no competing interests.

## NISC Comparative Sequencing Program

Beatrice B. Barnabas[2], Gerard G. Bouffard[2], Shelise Y. Brooks[2], Holly Coleman[2], Lyudmila Dekhtyar[2], Xiaobin Guan[2], Joel Han[2], Shi-ling Ho[2], Richelle Legaspi[2], Quino L. Maduro[2], Catherine A. Masiello[2], Jennifer C. McDowell[2], Cassandra Montemayor[2], Morgan Park[2], Nancy L. Riebow[2], Karen Schandler[2], Chanthra Scharer[2], Brian Schmidt[2], Christina Sison[2], Sirintorn Stantripop[2], Pamela J. Thomas[2], Meghana Vemulapalli[2] & Alice C. Young[2]

