## [Peer Review File · Nature Communications]

Reviewer #1 (Remarks to the Author):

The authors set out to characterize effects of genetics, sex, and 'life experience' on hepatic DNA methylation in mice. They studied males and females of two different inbred strains of mice (C57 and C3H) as well as F1 hybrids produced from crossing these two strains. They performed genome-wide sequencing to identify genomic variants between C57 and C3H mice, and genome-wide bisulfite sequencing to assess hepatic DNA methylation for all mice. The data sets appear to be of high quality, and appropriate analyses were performed to identify genotype-specific and sex-specific differentially methylated regions (DMRs). In my opinion, however, important issues related to study design and data interpretation reduce the likely impact of these studies.

Major concerns

There is a major flaw in study design. The authors studied primiparous C57 and C3H females, but virgin F1 hybrid females. During pregnancy in the mouse, the maternal liver undergoes dramatic remodeling, including a doubling in size, extensive increases in hepatocyte ploidy, and major changes in gene expression. Hence, it is likely that the cellular composition of the liver of virgin females is quite different from that of primiparous females, even some time after pregnancy. (I don't believe the authors ever state the age at which the mice were killed.) And, as the authors suggest, pregnancy may induce major changes in epigenetic regulation within specific cell types in the liver. Unfortunately, the focus of the paper is on genetic effects on hepatic DNA methylation (comparing, for example, methylation in F1 hybrids vs. pure C57). Also, in their sex comparisons they lump together all the females (some of which are virgin and some of which are primiparous). And, in their analysis of 'life experience' effects, pregnancy is completely confounded with genotype, so it is not clear how effects of pregnancy can be distinguished from effects of genotype in females.

The other major question is related, and focuses on the question of interpretation. The authors chose to study methylation in liver, but liver is a heterogeneous tissue, comprised predominantly of hepatocytes, endothelial cells, Kupffer cells, and Ito cells. So, in the studies comparing different genotypes, do the authors believe their methylation data reflect strain-specific differences in the cellular composition of the liver, or genetic effects that alter methylation in a subset (or majority) of the different hepatic cell types? (Notably, their finding that TF motifs associated with allele-specific methylation are enriched for associations with liver development and differentiation seem consistent with the former.) It would have been helpful to additionally study other (non-hepatic) tissues to help clarify this question.

The authors state in the manuscript that all the data have been deposited in GEO. Various processed data files are deposited, but I could not find the raw read data (SRA). Authors must make publicly available all raw read data (sequencing and bisulfite sequencing).

More minor concerns

The tables of DMRs are not informative. They should include annotation of each DMR with distances relative to genes, numbers of CpGs/DMR, average methylation in both genotypes, p value of differential methylation, read depth in each genotype, etc.

In several places, the authors refer to the genetic comparisons as indicating “methylation inheritance”. This is not correct. Their data indicate genetic influences on methylation.

Reviewer #2 (Remarks to the Author):

This manuscript describes the generation of 24 somatic tissue (liver) methylomes obtained from individuals from crosses of two inbred mouse strains in order to understand the DNA methylation changes across generations and between sexes. The analysis appears sound and many observations make sense in light of previous work (for example, DNA methylation is rather stable across individuals, and many differentially methylated regions tend to be far from CGIs etc). The data is of good quality and would be a valuable resource to researchers interested in epigenetics and methylation variation. I would recommend this study for publication.

Minor comments:

Many of the DMRs are enriched for Hox binding sites – it would be good to note that even though the canonical Hox binding site does not contain a CG dinucleotide, that this family of TFs can bind methylated DNA (PMID: 28473536)

Reviewer #3 (Remarks to the Author):

The present manuscript reports the effects of genotype, sex, and the life experience of pregnancy and/or lactation on the somatic methylomes of mouse liver tissue. The liver methylomes were determined from inbred mouse strains C57BL/6N and C3H/HeN (n=3 of each) as well as three F1 progeny from 6 total inter-strain crosses in both directions (i.e., cross 3 males and 3 females of each strain). Technically, the work is very sound in that bisulfite sequencing was performed on two or three separate libraries for three different animals in each experimental group to identify differentially methylated regions (DMRs). DMRs were identified based on strain (C57BL/6N versus C3H/HeN) and male versus female, i.e., sex DMRs.

A main finding is that DNA methylation is closely linked to DNA sequence, namely, alterations in the binding motifs of a subset of transcription factors expressed in liver are causally linked to allele-specific changes in DNA methylation. Using electromobility gel shift assays, the authors demonstrate that higher affinity binding sites in DMRs for the transcription factor FoxA1 correlate with lower levels of DNA methylation. As in vitro binding may not reflect in vivo binding, testing the differential binding of FoxA1 to least two of the motifs in C57BL/6N and C3H/HeN mice (testing F1 progeny is not needed) would offer critical support to the paper. A striking finding is that the patterns of DNA methylation were stably associated with DNA sequence regardless of passing through the female or male germ line. Overall, the paper reports important findings that add to a growing body of literature showing that transcription factors can set DNA methylation state. The major claims are novel, well supported, and should be of broad interest to the epigenetic community.

Major points:

1. Line 83: "Shotgun whole-genome bisulfite sequencing was performed essentially as described (ref. 25)." In order for readers to replicate the findings, it is important that the authors either verify or give more experimental detail, indicating which steps in the procedure were different from Lister et al. (ref. 25). In particular, Lister et al. does not indicate what Phred score was used to filter reads. The authors have used Q20, which would increase the number of obtained reads; however, my understanding is that improvements in sequencing beyond those available to Lister et al. have made the higher Q30 standard in the field.

It is unlikely that single-end reads of 87 cycles were obtained as in Lister et al. Was paired-end sequencing performed and for how many cycles?

What were the error rates in sequencing and non-conversion of lambda sequences? Reporting the highest rate of all samples should suffice.

As in Lister et al., were “reads that contained more than 3 cytosines in a non-CG context” used as a metric “to remove reads that were likely not bisulfite converted” or was another criterion used?

2. What are the scatter plot correlation coefficients between the two to three libraries that were constructed for each sample?

3. The genome browser views of Fig. 2A and Suppl. Fig 2A appear to be identical. I think this would be expected as both the DSS and Metilene programs call DMRs but the methylation levels at each CpG site are the same. It would be more informative to show genomic regions that were assigned as DMRs by one program but not the other.

4. Suppl. Table S3: “Both computational methods called DMRs of each ‘polarity’ (BL/6 methylation level higher than C3H and vice versa) at equivalent rates (Table S3).” Adding the total number of DMRs of each polarity as a footnote in the spreadsheet (or elsewhere) for each computational method would allow readers to quickly assess the veracity of this statement. It would also be helpful to split the DMRs by direction and indicate their rank methylation scores in the table to correspond with Figure 3A and Suppl. Fig. 3A.

5. A Venn Diagram showing the overlap and non-overlap of DMRs obtained by DSS versus Metilene would be useful.

6. Line 191: Ooi et al. Nature. 2007. 448:714 should be cited as a mechanism by which chromatin affects DNA methylation.

7. Fig. 4D-H: It would be very helpful to label each motif as C3>B6 (Fig. 4D-F) or B6>C3 (Fig. 4G-H) to allow readers to quickly evaluate the statement on lines 199-200 (instead of needed to search for each element in Suppl. Table S3).

8. Lines 259-261: “Like the genotype-DMRs, sex-DMR-enriched motifs were not enriched in sequences containing CpG dinucleotides.” Given that five of the genotype-DMRs contained CpGs, one could argue that there is partial enrichment for CpGs. Also, two of the motifs in sex-DMRs (HNF6 and CEBP) contain CpGs.

Minor comments:

1. Line 77: refer to C57BL/6N and C3H/HeN as B6 and C3, respectively, to agree with the labels in Suppl. Fig. 1 and for consistency throughout the manuscript?

2. Line 318: change 1-20 to 1-10.

3. Line 321: "CpG sites are excluded if <18 animals have depth >10" is unclear.

4. Line 328: Indicate that the C3>B6 DMR bar is green.

5. Line 345: It is obvious; however, indicate that the DMR is B6>C3.

6. Line 180: instead of 18, 20 transcription factor motifs were found to be enriched by Homer in Figure 4A-B.

7. Line 196 and 198: "...asked whether SNVs within DMRs can interfere with DNA binding in a biochemical assay." The wording should be changed to "SNVs within or adjacent to DMRs", because the FoxA1 motifs in Figs. 4H and 4I are both located 20 bp outside of their DMRs defined in

8. Figure 2C: Capitalize the Ks.

9. Figure 3A, 6A, Suppl. 3A, and Suppl. 6A: histogram is misspelled and the words are masked by the color key bar.

10. Figure 4D-F: Again, it is obvious, but would showing the reverse complement sequences facilitate readers quickly identifying the FoxA1 motif. That stated, I can see why the authors may purposely want to show the two different strands.

Response to review

We thank the editorial staff at Nature Communications and the three peer reviewers for the comments on our manuscript. We found the comments to be insightful. We believe that our responses have substantially improved the manuscript.

Reviewer #1 (Remarks to the Author):

The authors set out to characterize effects of genetics, sex, and 'life experience' on hepatic DNA methylation in mice. They studied males and females of two different inbred strains of mice (C57 and C3H) as well as F1 hybrids produced from crossing these two strains. They performed genome-wide sequencing to identify genomic variants between C57 and C3H mice, and genome-wide bisulfite sequencing to assess hepatic DNA methylation for all mice. The data sets appear to be of high quality, and appropriate analyses were performed to identify genotype-specific and sex-specific differentially methylated regions (DMRs). In my opinion, however, important issues related to study design and data interpretation reduce the likely impact of these studies.

We thank the reviewer for commenting positively on the quality of our data and for supporting our choice of analyses to identify differentially methylated regions.

Major concerns

There is a major flaw in study design. The authors studied primiparous C57 and C3H females, but virgin F1 hybrid females. During pregnancy in the mouse, the maternal liver undergoes dramatic remodeling, including a doubling in size, extensive increases in hepatocyte ploidy, and major changes in gene expression. Hence, it is likely that the cellular composition of the liver of virgin females is quite different from that of primiparous females, even some time after pregnancy. (I don't believe the authors ever state the age at which the mice were killed.) And, as the authors suggest, pregnancy may induce major changes in epigenetic regulation within specific cell types in the liver. Unfortunately, the focus of the paper is on genetic effects on hepatic DNA methylation (comparing, for example, methylation in F1 hybrids vs. pure C57). Also, in their sex comparisons they lump together all the females (some of which are virgin and some of which are primiparous). And, in their analysis of 'life experience' effects, pregnancy is completely confounded with genotype, so it is not clear how effects of pregnancy can be distinguished from effects of genotype in females.

This is an interesting comment and one that gave us pause for thought.

We apologize that the details of animal husbandry (including timing of pregnancy and weaning relative to sacrifice) were not available in the manuscript. We now include this information in the Methods section. In short, animals were mated at 10 weeks of age. Pregnant females delivered at week 13 and pups were weaned at week 16. All animals were sacrificed at week 20 regardless of parity.

The reviewer suggests a confounding effect of parity status in our design. We respectfully disagree. In our comparison across genotype, we compared 3 male C57BL/6 animals and 3 primiparous C57BL/6 females to 3 male C3H/He animals and 3 primiparous C3H/He animals. Any impact on methylation pattern contributed by parity is balanced across this comparison with three primiparous animals per group.

To address the cellular status of livers of the animals in our study, we retrieved the paraffin blocks prepared from the right liver lobe of all animals in the study. Sections were cut and stained with H&E

for analysis. Two trained scientists screened the sections for general appearance. We observed similar histologic features for all animals in the study (regardless of sex, genotype or parity). We likewise observed similar numbers of cells per field for all animals (counting number of nuclei in 10 fields per animal for all 24 animals in the study), indicating that at the time of sacrifice cell size in the dams had returned to baseline. Representative and summary data are now presented in Supplementary Materials.

Finally, we note that our comparison of C57BL/6 animals to C3H/He animals resulted in identification of thousands of DMRs which, when quantitated and displayed across all animals (including the nulliparous F1 animals that were not part of the comparison used to define DMRs— see Figure 3A, Supplemental Figure 3A) have DNA methylation levels in the F1 animals intermediate to that of the parental genotypes. Thus, the data do not behave as though there is any confounding as suggested by the reviewer. In all cases, nulliparous animals have a methylation level intermediate to the levels of the two parental genotypes. We concluded that the parity status of the female animals was not a major confounder. Given this additional analysis, we believe that our design to explore the impact of genotype on liver methylation is not confounded by parity status.

Regarding the comparison by sex, the reviewer is mistaken. We did not lump together all female animals and compare to all male animals. We compared all dams (both C3H/H3 and C57BL/6) to all sires – no progeny were included in the data for DMR calculation. In the visualization presented in Figure 6, all animals were presented in the heatmap. We have now clearly indicated this fact in the text. We do observe an impact of parity in this comparison – and we go on to identify the the apparent cause of this confounder as the impact of life experience in this comparison.

Regarding the reviewer's comments on the confounding impact of genotype in the comparison of dams to daughters, we once again respectfully disagree. This analysis compares all 6 dams with all 6 virgin female offspring. There are an exactly equal number of C3H/He and C57BL/6 alleles in each group at each locus (6 total per group). We fail to see how genotype can be a confounder in this circumstance.

The other major question is related, and focuses on the question of interpretation. The authors chose to study methylation in liver, but liver is a heterogeneous tissue, comprised predominantly of hepatocytes, endothelial cells, Kupffer cells, and Ito cells. So, in the studies comparing different genotypes, do the authors believe their methylation data reflect strain-specific differences in the cellular composition of the liver, or genetic effects that alter methylation in a subset (or majority) of the different hepatic cell types? (Notably, their finding that TF motifs associated with allele-specific methylation are enriched for associations with liver development and differentiation seem consistent with the former.) It would have been helpful to additionally study other (non-hepatic) tissues to help clarify this question.

This is also an interesting comment/question. We have performed histologic analyses across all animals in our study and found no visible difference in cell type composition of the tissue. We believe the data (as suggested by the TF motif analysis) reflect differences in hepatocyte DNA methylation levels at the indicated loci. Representative data are now in Supplementary Materials. We agree that additional methylomes could clarify this question, we respectfully suggest that further sequencing and analysis is beyond the scope of the current manuscript.

The authors state in the manuscript that all the data have been deposited in GEO. Various processed data files are deposited, but I could not find the raw read data (SRA). Authors must make publicly available all raw read data (sequencing and bisulfite sequencing).

The raw read data are deposited at SRA. We do not have the ability to make this visible to reviewers, but the GEO data series page that is available to reviewers clearly indicates that primary data are available at SRA. We are happy to release the sequencing data (including both bisulfite and DNA sequencing as well as the new ChIP sequencing) when the manuscript is accepted for publication – it is deposited as we previously stated.

More minor concerns

The tables of DMRs are not informative. They should include annotation of each DMR with distances relative to genes, numbers of CpGs/DMR, average methylation in both genotypes, p value of differential methylation, read depth in each genotype, etc.

We have added the requested information in all DMR tables.

In several places, the authors refer to the genetic comparisons as indicating “methylation inheritance”. This is not correct. Their data indicate genetic influences on methylation.

We agree with this comment and have removed the term methylation inheritance.

Reviewer #2 (Remarks to the Author):

This manuscript describes the generation of 24 somatic tissue (liver) methylomes obtained from individuals from crosses of two inbred mouse strains in order to understand the DNA methylation changes across generations and between sexes. The analysis appears sound and many observations make sense in light of previous work (for example, DNA methylation is rather stable across individuals, and many differentially methylated regions tend to be far from CGIs etc). The data is of good quality and would be a valuable resource to researchers interested in epigenetics and methylation variation. I would recommend this study for publication.

We thank this reviewer for the favorable comments on our work.

Minor comments:

Many of the DMRs are enriched for Hox binding sites – it would be good to note that even though the canonical Hox binding site does not contain a CG dinucleotide, that this family of TFs can bind methylated DNA (PMID: 28473536)

We thank the reviewer for this comment. We have added appropriate language to the text to reflect this observation.

Reviewer #3 (Remarks to the Author):

Wade NatCommun Genetics Sex & Life Experience Effects on DNA Methylation 150578 Review 01142018

The present manuscript reports the effects of genotype, sex, and the life experience of pregnancy and/or lactation on the somatic methylomes of mouse liver tissue. The liver methylomes were determined from inbred mouse strains C57BL/6N and C3H/HeN (n=3 of each) as well as three F1 progeny from 6 total inter-strain crosses in both directions (i.e., cross 3 males and 3 females of each

strain). Technically, the work is very sound in that bisulfite sequencing was performed on two or three separate libraries for three different animals in each experimental group to identify differentially methylated regions (DMRs). DMRs were identified based on strain (C57BL/6N versus C3H/HeN) and male versus female, i.e., sex DMRs.

We thank this reviewer for noting the care we have taken to ensure the data presented here are of the highest quality we can achieve.

A main finding is that DNA methylation is closely linked to DNA sequence, namely, alterations in the binding motifs of a subset of transcription factors expressed in liver are causally linked to allele-specific changes in DNA methylation. Using electromobility gel shift assays, the authors demonstrate that higher affinity binding sites in DMRs for the transcription factor FoxA1 correlate with lower levels of DNA methylation. As in vitro binding may not reflect in vivo binding, testing the differential binding of FoxA1 to at least two of the motifs in C57BL/6N and C3H/HeN mice (testing F1 progeny is not needed) would offer critical support to the paper. A striking finding is that the patterns of DNA methylation were stably associated with DNA sequence regardless of passing through the female or male germ line. Overall, the paper reports important findings that add to a growing body of literature showing that transcription factors can set DNA methylation state. The major claims are novel, well supported, and should be of broad interest to the epigenetic community.

We thank the reviewer for the enthusiastic description of our work and conclusions. We also believe this work is of broad interest to the epigenetic community. As the reviewer suggested, we conducted Foxa1 ChIP-seq using B6 and C3 livers, and confirmed that DMR-associating SNVs found within Foxa1 motifs impact FOXA1 binding in liver. This new data is presented in the main and supplementary figures.

Major points:

1. Line 83: "Shotgun whole-genome bisulfite sequencing was performed essentially as described (ref. 25)." In order for readers to replicate the findings, it is important that the authors either verify or give more experimental detail, indicating which steps in the procedure were different from Lister et al. (ref. 25). In particular, Lister et al. does not indicate what Phred score was used to filter reads. The authors have used Q20, which would increase the number of obtained reads; however, my understanding is that improvements in sequencing beyond those available to Lister et al. have made the higher Q30 standard in the field.

We thank the reviewer for this comment. As the reviewer suspected, we have neglected to sufficiently describe the methods used. We have now added additional detail to the Methods section to address how our data generation was performed. We agree with this reviewer that the use of newer instrumentation with patterned flow cells greatly increases the quality of bisulfite sequencing data. Our data were collected on the HiSeq2000 platform. This platform has some issues with cluster identification and calling for bisulfite sequencing due to the loss of information following conversion. This is particularly evident in read 2 where cluster size expands resulting in lower Phred scores. Phred score of 20 is, we feel, very reasonable given the platform utilized.

It is unlikely that single-end reads of 87 cycles were obtained as in Lister et al. Was paired-end sequencing performed and for how many cycles?

The reviewer is correct. We performed PE100 base sequencing. This is now stated in methods.

What were the error rates in sequencing and non-conversion of lambda sequences? Reporting the highest rate of all samples should suffice.

Bisulfite conversion rates were almost always in excess of 99.5%. Values for each library are given in Table S1.

As in Lister et al., were “reads that contained more than 3 cytosines in a non-CG context” used as a metric “to remove reads that were likely not bisulfite converted” or was another criterion used?

Given the high rate of conversion we observed, we did not use this type of filter.

2. What are the scatter plot correlation coefficients between the two to three libraries that were constructed for each sample?

These values are now supplied in Table S2

3. The genome browser views of Fig. 2A and Suppl. Fig 2A appear to be identical. I think this would be expected as both the DSS and Metilene programs call DMRs but the methylation levels at each CpG site are the same. It would be more informative to show genomic regions that were assigned as DMRs by one program but not the other.

Agreed. We have modified the figures where appropriate to depict DMRs assigned by one program but not the other.

4. Suppl. Table S3: “Both computational methods called DMRs of each ‘polarity’ (BL/6 methylation level higher than C3H and vice versa) at equivalent rates (Table S3).” Adding the total number of DMRs of each polarity as a footnote in the spreadsheet (or elsewhere) for each computational method would allow readers to quickly assess the veracity of this statement. It would also be helpful to split the DMRs by direction and indicate their rank methylation scores in the table to correspond with Figure 3A and Suppl. Fig. 3A.

The DMR polarity numbers are available in Table 1. Rank ordering or splitting by directionality is trivial in Excel, we presume readers can sort the data by whatever criteria they wish.

5. A Venn Diagram showing the overlap and non-overlap of DMRs obtained by DSS versus Metilene would be useful.

Agreed. See figure S2.

6. Line 191: Ooi et al. Nature. 2007. 448:714 should be cited as a mechanism by which chromatin affects DNA methylation.

Agreed. Done.

7. Fig. 4D-H: It would be very helpful to label each motif as C3>B6 (Fig. 4D-F) or B6>C3 (Fig. 4G-H) to allow readers to quickly evaluate the statement on lines 199-200 (instead of needed to search for each element in Suppl. Table S3).

Agreed. We have indicated on the figure and in the legend which allele bears more methylation.

8. Lines 259-261: "Like the genotype-DMRs, sex-DMR-enriched motifs were not enriched in sequences containing CpG dinucleotides." Given that five of the genotype-DMRs contained CpGs, one could argue that there is partial enrichment for CpGs. Also, two of the motifs in sex-DMRs (HNF6 and CEBP) contain CpGs.

This is a fair comment. We have deleted this sentence. Readers can judge for themselves.

Minor comments:

1. Line 77: refer to C57BL/6N and C3H/HeN as B6 and C3, respectively, to agree with the labels in Suppl. Fig. 1 and for consistency throughout the manuscript?

Agreed. Done.

2. Line 318: change 1-20 to 1-10.

Thank for catching this. Correction made.

3. Line 321: "CpG sites are excluded if <18 animals have depth >10" is unclear.

Agreed. We have modified the statement. Essentially, if not enough animals have sufficient read depth at a given CpG site, we have excluded it from analysis.

4. Line 328: Indicate that the C3>B6 DMR bar is green.

Agreed. Done.

5. Line 345: It is obvious; however, indicate that the DMR is B6>C3.

Agreed. Done.

6. Line 180: instead of 18, 20 transcription factor motifs were found to be enriched by Homer in Figure 4A-B.

Thanks for catching this. Corrected.

7. Line 196 and 198: "...asked whether SNVs within DMRs can interfere with DNA binding in a biochemical assay." The wording should be changed to "SNVs within or adjacent to DMRs", because the FoxA1 motifs in Figs. 4H and 4I are both located 20 bp outside of their DMRs defined in

Agreed. Thank for catching this. We have corrected the statement.

8. Figure 2C: Capitalize the Ks.

K's are now capitalized.

9. Figure 3A, 6A, Suppl. 3A, and Suppl. 6A: histogram is misspelled and the words are masked by the color key bar.

Thanks for catching this. We have corrected the figures.

10. Figure 4D-F: Again, it is obvious, but would showing the reverse complement sequences facilitate readers quickly identifying the FoxA1 motif. That stated, I can see why the authors may purposely want to show the two different strands.

To facilitate clarity, we have aligned the motif over the strand sequence (using the appropriate strand information to match the consensus).

Reviewer #1 (Remarks to the Author):

I am satisfied with the authors' clarifications in response to my concerns, and the additional data included in the manuscript.

Reviewer #2 (Remarks to the Author):

The reviewers have satisfactorily addressed my comments.

Reviewer #3 (Remarks to the Author):

I stand by my original, positive assessments of the manuscript, and all of my comments have been satisfactorily addressed. The FoxA1 ChIP-seq results showing that in vivo site occupancies correlate well with biochemical affinity of the DNA-binding domain are an excellent addition. The manuscript will be well received by researchers working in the areas of chromatin biology and epigenetics.

National Institute of
Environmental Health Sciences

Paul A. Wade, Ph.D.
Senior Investigator and Deputy Chief
Epigenetics and Stem Cell Biology Laboratory
NIEHS, National Institutes of Health
111 TW Alexander Drive, Bldg. 101/D416A
Research Triangle Park, NC 27709
Email: wadep2@mail.nih.gov
Phone: (984) 287-4133; Fax (301) 480-2989

October 31, 2018

To: Dr. Carolina Perdigoto, Ph.D.
Senior Editor, Nature Communications
Re: Response to reviews, manuscript NCOMMS-17-31339A

Dear Dr. Perdigoto,

We thank the reviewers for their endorsement of our work.

REVIEWERS' COMMENTS:

Reviewer #1 (Remarks to the Author):

I am satisfied with the authors' clarifications in response to my concerns, and the additional data included in the manuscript.

Reviewer #2 (Remarks to the Author):

The reviewers have satisfactorily addressed my comments.

Reviewer #3 (Remarks to the Author):

I stand by my original, positive assessments of the manuscript, and all of my comments have been satisfactorily addressed. The FoxA1 ChIP-seq results showing that in vivo site occupancies correlate well with biochemical affinity of the DNA-binding domain are an excellent addition. The manuscript will be well received by researchers working in the areas of chromatin biology and epigenetics.

Thank you for an instructive and useful scientific interaction with the reviewers.

Sincerely,

Paul Wade